# Body Mass Index Mediates the Relationship between the Frequency of Eating Away from Home and Hypertension in Rural Adults: A Large-Scale Cross-Sectional Study

**DOI:** 10.3390/nu14091832

**Published:** 2022-04-27

**Authors:** Beibei Liu, Xiaotian Liu, Yuyang Wang, Xiaokang Dong, Wei Liao, Wenqian Huo, Jian Hou, Linlin Li, Chongjian Wang

**Affiliations:** Department of Epidemiology and Biostatistics, College of Public Health, Zhengzhou University, Zhengzhou 450001, China; bbliu2018@163.com (B.L.); xtliu2008@126.com (X.L.); wyuyang2018@163.com (Y.W.); dong4568529173@163.com (X.D.); wliaotr@163.com (W.L.); huowenqian@zzu.edu.cn (W.H.); houjian1988@zzu.edu.cn (J.H.); lilinlin@zzu.edu.cn (L.L.)

**Keywords:** eating away from home, blood pressure, hypertension, mediation analysis, rural population

## Abstract

This study was conducted to investigate the association of eating away from home (EAFH) frequency with hypertension and to explore whether the association was mediated by BMI. A total of 29,611 participants were selected from the Henan Rural Cohort Study. Data on the frequency of EAFH were obtained by face-to-face questionnaires. The relationship between EAFH frequency and blood pressure was evaluated by linear regression. Logistic regression and restricted cubic spline were used to assess the association between EAFH frequency and hypertension, and the mediation effect of BMI on the relationship was performed. There were pronounced associations between the frequency of EAFH and blood pressure (*P* *trend* < 0.001) in the total population and men. Compared with the population with 0 times EAFH per week, the multivariate odds ratios (*OR*s) and 95% confidence intervals (95% *CI*s) for hypertension of the group with 7 times or more EAFH per week were 1.673 (1.482–1.889) for the total population and 1.634 (1.413–1.890) for men. A nonlinear dose-response relationship was detected between the frequency of EAFH and hypertension (*p* < 0.001), and the relationship was partially mediated by BMI. The proportion explained was 21.3% in the total population and 25.4% in men. The current study indicated that EAFH was associated with rising blood pressure and increased risk of hypertension and BMI partially mediated the relationship.

## 1. Introduction

Hypertension is one of the most common chronic noncommunicable diseases in the world. According to the World Health Organization, an estimated 1.28 billion adults aged 30–79 years worldwide suffer from hypertension, of whom about two-thirds live in low- and middle-income countries. Data from around the world have indicated that the prevalence of hypertension among people aged 30–79 years has remained stable globally over the past three decades (from 1990 to 2019) (32% vs. 32% in women, 32% vs. 34% in men) [1]. According to the National Health Commission of China, the prevalence of hypertension among people aged 18 and over in China was 25.20% in 2012 and increased to 27.90% in 2015. The estimated prevalence of hypertension in China was 31.89% in 2019. The growth rate of hypertension in China is higher than that at the world level. However, the treatment and control rate of hypertension is relatively low [2]. The most worrying thing is that many complications occur with the continuous development of hypertension. The familiar complications are cardiovascular disease, stroke, kidney disease, and so on [3,4,5]. Many studies have shown that risk factors for hypertension include genetics, overweight and obese status, high-salt diet, hyperlipidemia, hyperglycemia, and unhealthy living habits (smoking, drinking, etc.) [6,7,8]. Therefore, intervention in these risk factors can effectively prevent hypertension and reduce the prevalence of hypertension.

Currently, more and more people are choosing to eat away from home, and the frequency of EAFH (eating away from home) is increasing year by year worldwide [9,10]. Chinese residents, including urban and rural people, are no exception and are spending more and more on EAFH. According to the National Bureau of Statistics, the proportion of catering revenue in the national food, tobacco, and alcohol consumption expenditure increased from 45.5% in 2013 to 52.9% in 2017, and it is expected to reach 61.0% in 2023, which may indicate an increase in the frequency of EAFH. However, many studies have depicted that EAFH always involves consuming more energy, fat, salt, and sugar [11,12]. According to these characteristics and the risk factors of hypertension, it can be inferred that EAFH may cause elevated blood pressure or even hypertension. However, previous studies have explored the relationship of EAFH with obesity, diabetes, and so on. No studies were found to examine the relationship between the frequency of EAFH and hypertension, especially in the rural population, which accounts for about two-fifths of the total population in China. Compared with the urban population, the prevalence of hypertension in the rural population is higher, but the treatment and control rates are lower [13]. Thus, the prevention of hypertension in the rural population can remarkably lessen the burdens on the nation and on families and would be of great significance in reducing the overall prevalence of hypertension.

Generally speaking, the relationship between the frequency of EAFH and hypertension remains unclear. The purpose of this study was to explore this relationship based on the rural adults of Henan Province and to assess whether BMI plays a mediating role in the association between the frequency of EAFH and hypertension.

## 2. Materials and Methods

### 2.1. Study Population

The research subjects came from the Henan Rural Cohort Study, which began in July 2015 in Suiping, Yuzhou, Tongxu, Xinxiang, and Yima counties. The subjects were selected from the general population aged 18 to 79 by a multistage cluster sampling method. Detailed information was described in a previous paper [14]. Ultimately, 39,259 people (15,490 men and 23,769 women) participated in the study. If participants had cancer (n = 291) or severe kidney disease (n = 18) or lost data on the frequency of EAFH (n = 9291) or blood pressure or hypertension (n = 48), they were no longer subjects of this study. Ultimately, 29,611 participants were retained for the current study.

The protocol of this study was approved by the ethics committee of the Zhengzhou University Life Science Ethics Committee. Informed consent was obtained from all participants.

### 2.2. The Measurement of Blood Pressure and the Definition of Hypertension

Participants were told not to smoke, exercise, or drink coffee within 30 min before blood pressure was measured. Upon arrival at the measuring site, the participants sat down and rested for five minutes and then measured the blood pressure on their right arm. The blood pressure was measured 3 times with an electronic sphygmomanometer (HEM-770AFuzzy, Omron, Japan), and then participants had a break of at least 30 s after each measurement. The mean of the three measurements was used for this study.

Hypertension was defined as the mean of 3 measurements of blood pressure, SBP ≥ 140 mmHg and/or DBP ≥ 90 mmHg. In addition, people who had been diagnosed with hypertension and were taking antihypertensive drugs were included [15].

### 2.3. Definition and Assessment of EAFH

EAFH was defined according to the source of food. As long as the food was bought outside the home and not made at home, it was EAFH, no matter where the food was eaten. EAFH included the food from both dine-in and takeout restaurants, as well as fast food shops, food courts, food stands, or grocery stores [16,17].

Through a face-to-face questionnaire survey carried out by well-trained investigators, relevant data on the frequency of EAFH were acquired. The questions included: “How many times have you eaten breakfast, lunch, or dinner in the past week?” and “How many times have you eaten away from home for breakfast, lunch, and dinner in the past week?” The frequency of EAFH weekly was divided into 0, 1–2, 3–4, 5–6, and ≥7 times.

### 2.4. Assessment of Covariates

General demographic characteristics, lifestyle, and personal and family history of disease were obtained by face-to-face questionnaires. Marital status was sorted into married/cohabiting and unmarried/divorced/widowed. Education level was divided into junior high school or below, high school, and high school above. The average monthly income of each person was classified into <500 CNY, 500–1000 CNY and ≥1000 CNY. Smoking status was categorized into three types: never, former (ever smoked), and current (smoked for at least six months and at least one cigarette per day). Alcohol consumption was grouped into three categories: never, former (ever drank), and current (drank 12 or more times a year). According to the International Physical Activity Questionnaire (IPAQ) [18], physical activity was classified into low, moderate, and high. Abundant vegetable and fruit intake referred to eating no less than 500 g of vegetables and fruits every day, and high-fat diet was defined as the average daily consumption of more than 75 g of livestock and poultry meat [19]. Family history of hypertension was defined as a parent or sibling having a history of hypertension.

When measuring height and weight, participants were asked to wear light clothes and no shoes. Moreover, the measurements of height and weight were accurate to 0.1 cm and 0.1 kg. Body mass index (BMI) was calculated as weight divided by height squared (kg/m^2^).

### 2.5. Statistical Analysis

For the basic characteristics of the research object, the continuous variables are expressed as means ± standard deviations and were analyzed by Student’s *t*-test, and categorical variables are shown in quantities (percentages) and were analyzed using the chi-squared test. Multiple linear regression was used to examine the relationship between EAFH frequency and blood pressure according to correlation coefficients (*β*) and 95*%* confidence intervals. The association between the frequency of EAFH and hypertension was determined by the odds ratios (*OR*s) and the 95*%* confidence intervals via binary logistic regression. The dose–response relationship between continuous EAFH frequencies and hypertension was detected by restricted cubic spline in logistic regression. Three models were established in this study: Model 1—adjusted for age; Model 2—adjusted for age, gender (only for total participants), marital status, education level, and average income per month; Model 3—adjusted for age, gender (only for total participants), marital status, education level, average income per month, smoking status, alcohol consumption, physical activity, abundant vegetable and fruit intake, high-fat diet, and family history of hypertension. The mediation analysis was completed by applying PROCESS for SPSS [20,21] to explore the mediating effect of BMI on the relationship between the frequency of EAFH and hypertension. The mediating effect of BMI was obtained by the ratio of the indirect effect to the total effect. Partial mediation referred to the occasion when the frequency of EAFH and hypertension had apparent direct and indirect effects.

All data were run twice and analyzed by SPSS21.0 (Chicago, IL, USA) and R version 3.63. Results with *p* < 0.05 were considered to have statistical significance.

## 3. Results

### 3.1. Characteristics of Study Populations

Table 1 reveals the basic characteristics of the participants. The average age of the participants was 55.4 ± 12.4 years. There were 3892 men and 5593 women with hypertension. The participants with hypertension had the following characteristics: older age; higher BMI; lower education level, ratios of smoking and drinking, income, and physical activity; and less intake of vegetables and fruits. The frequency of EAFH was statistically significant with whether subjects suffered from hypertension among the total population (*p* < 0.001).

### 3.2. Association between the Frequency of EAFH and Blood Pressure and Hypertension

Figure 1 and Appendix A Appendix A portray the relationship between the frequency of EAFH and blood pressure (SBP and DBP). Figure 1 was made according to Appendix A. It could be found that, in the three models, there were positive associations of EAFH frequency with SBP and DBP (*P trend* < 0.001) in the total population and men but not in women. In model 3, compared with the reference group, when the frequency of EAFH was ≥7 times, the total population, men, and women had the highest β-coefficients (95% *CI*) of DBP, which were 2.825 (2.252–3.398), 2.990 (2.262–3.718), and 0.880 (−0.067–1.827), respectively; for SBP, these were 3.931 (3.008–4.853) and 3.504 (2.380–4.627) in the total population and men, respectively, when the frequency of EAFH was ≥7 times and 2.035 (−1.303–5.374) for women when the frequency was 5–6 times. In addition, 1-SD increases in the frequency of EAFH were significantly associated with SBP increasing by 24.1% and 18.7%, and DBP increasing by 17.4% and 16.6%, in the total population and men, respectively.

The association between EAFH frequency and hypertension is shown in Table 2. In the three models, as the frequency of EAFH grew, the *OR* of the total population and men also gradually increased. In the fully adjusted model, the *OR* was the largest for the total population and men who ate away from home 7 times or more, at 1.673 (95% *CI* 1.482–1.889) and 1.634 (95% *CI* 1.413–1.890), respectively, and for women who ate away from home 5–6 times, at 1.393 (95% *CI* 0.800–2.426). In model 3, *OR*s and 95% *CI*s for each level increment in the frequency of EAFH were 1.031 (1.022–1.040), 1.026 (1.016–1.037), and 1.010 (0.993–1.027) in the general population, men, and women, respectively. Furthermore, there was a clear nonlinear dose-response relationship between the frequency of EAFH and hypertension in the total population and men (Figure 2).

### 3.3. Association between the Frequency of EAFH Breakfasts, Lunches, and Dinners and Hypertension

In the overall population, taking the participants with 0 times EAFH per week as the control group, those who ate away from home for breakfast, lunch, and dinner had an increased risk of hypertension, except when the frequency of EAFH for breakfast was 3–4 times per week (*p* > 0.05) (Figure 3). Figure 3 was made according to Appendix A Appendix A. After gender stratification, men who ate out at lunch had elevated odds of hypertension, and those who for whom the frequency of EAFH per week for breakfast was 5–6 times (*OR* 1.565, 95% *CI* 1.013–2.418) and for dinner was 3–4 times (*OR* 1.713, 95% *CI* 1.328–2.208) had the highest likelihood of hypertension. Furthermore, for the general population and for men, each SD increment in the frequency of EAFH for breakfast increased the *OR* of hypertension by 5.5% (95% *CI* 2.9–8.1%) and 4.3% (95% *CI* 1.3–7.5%); each increment for lunch increased the *OR* by 7.5% (95% *CI* 5.4–9.6%) and 7.4% (95% *CI* 4.9–9.9%); and each increment for dinner increased the *OR* by 6.6% (95% *CI* 4.0–9.3%) and 4.4% (95% *CI* 1.3–7.5%), respectively (Appendix A Appendix A).

### 3.4. Mediation Effects

It was detected that the frequency of EAFH was positively associated with BMI in the total population and men by linear regression (Appendix A Appendix A). Further investigations were performed to determine whether the relationship between the frequency of EAFH and hypertension was mediated by BMI. The total effect of frequency of EAFH on hypertension was significant (*OR* 1.031, 95% *CI*, 1.022–1.040, *p* < 0.001). The relationship between the frequency of EAFH and hypertension was affected by the direct effect of EAFH frequency (*OR* 1.026, 95% *CI* 1.017–1.035, *p* < 0.001) and the indirect effect mediated by BMI (*OR* 1.006, 95% *CI* 1.004–1.009, *p* < 0.05) (Figure 4). The parameter estimate was 21.3% in the total population and 25.4% in men, indicating that BMI played a partial mediating role (Appendix A Appendix A).

## 4. Discussion

This cross-sectional study found that 32.03% of the participants suffered from hypertension, of which 32.17% of men and 31.94% of women had hypertension in rural areas of Henan Province. A national survey demonstrated that the prevalence of hypertension was 29.6% (31.2% for men and 28.0% for women) [22]. In addition, studies involving 115 rural and urban communities and northeast China indicated that rural areas had higher morbidity of hypertension than cities [23,24]. These studies showed that there was possibly a higher prevalence of hypertension in rural areas of Henan Province. Therefore, it may be necessary and significant to conduct research to prevent and control of hypertension.

Positive associations were found between the frequency of EAFH and blood pressure and hypertension, but these relationships did not manifest in women. Compared with eating out zero times per week, EAFH for lunch and dinner elevated the risk of hypertension in the total population, no matter how many times they ate out. Moreover, EAFH at lunch increased the odds of hypertension for men. Furthermore, the behavior of EAFH was associated with elevated BMI in the general population and men by linear regression. In addition, the relationship between the frequency of EAFH and hypertension was mediated by BMI at proportions of 21.3% and 25.4% for the general population and men, respectively.

A previous survey showed that EAFH was associated with elevated blood pressure in middle-aged men, which was consistent with this study [25]. Furthermore, a study focused on university-going young adults indicated that EAFH was associated with prehypertension [26]. In addition, a study from India suggested that fast food and inactivity were risk factors for obesity and hypertension [27]. Moreover, the consumption of fried foods was associated with a higher risk of prehypertension, hypertension, overweight, and obesity [28,29]. However, few studies have mentioned that EAFH had no significant association with hypertension, and the frequency of EAFH was not related to blood pressure in women in the current study. This may be the reason that different studies had different racial, sample sizes, study areas, lifestyle and customs, covariates, and so on. Therefore, more prospective research is deserved to explore and confirm the association of EAFH frequency with blood pressure and hypertension.

There are several possible mechanisms to explain the relationship between the frequency of EAFH and hypertension. First, the food characteristics of EAFH are closely related to the occurrence and development of hypertension. Some studies have shown that food from restaurants typically contains more energy, fat, salt, and sugar but less calcium, vitamin C, and micronutrients [11,12,30]. A previous study showed that reducing the frequency of EAFH could reduce sodium intake [31], which is a significant risk factor for hypertension [32]. Furthermore, the intake of red meat, processed meat, and sugary drinks increase when EAFH; this is positively correlated with the risk of hypertension [33]. Eating out also tends to increase exposure to ultra-processed foods, such as sweet or salty snacks, ice cream, French fries, hamburgers, hot dogs, and so on [34], which increases the possibility of hypertension [35,36]. Second, larger food portions are usually offered while eating away from home. When a person is provided with more food than they need, their consumption exceeds their need [37]. The consumption of more food away from home means more calories, which is more likely to lead to overweightness or obesity, which in turn are risk factors for hypertension [38]. Third, EAFH can enhance the risk of some chronic noncommunicable diseases that promote the occurrence of hypertension. Several studies have shown that diabetes and hyperuricemia are strongly associated with the frequency of EAFH [16,17], and diabetes and hyperuricemia are risk factors for hypertension [39].

The most interesting finding of this study was that the relationship between the frequency of EAFH and blood pressure and hypertension existed only in men. The difference between the two genders may be due to the following reasons. First, estrogen has a protective effect on women. Many studies have shown that estrogen has numerous benefits, including vasodilation, sympathetic inhibition, prevention of vascular remodeling, and so on [40,41], that reduce the risk of hypertension to a certain extent. Conversely, women have a more pronounced increase in blood pressure and have higher hypertension morbidity after menopause [42]. Of course, estrogen has adverse effects on women, such as increasing the risk of uterine fibroids and breast cancer [43]. Second, there are genetic differences between the sexes. Women have lower expression of angiotensin II type 1 receptor, angiotensin-converting enzyme, and plasma renin than men [44]. The alleles rs2074192 and rs2106809 of ACE2 play important roles in lowering circulating Ang (1–7) in women with hypertension [45]. Third, compared with men, women have a healthier lifestyle. According to a previous study, men are more likely than women to drink heavily (76% vs. 62%) and smoke (34% vs. 27%) [46]. In addition, men and women have different food preferences when EAFH, with men preferring high-fat diets and women preferring low-carbohydrate diets [25]. Furthermore, alcohol is often consumed by men when dining out, which may increase blood pressure by stimulating the renin–angiotensin–aldosterone system [47]. Moreover, influenced by traditional Chinese thought, men have more social activities. Therefore, men have more opportunities to eat out.

Some studies have shown that EAFH is associated with obesity [48,49], which is consistent with the finding of this study that BMI increased while eating out. This result may be related to the characteristics of EAFH, namely high energy and large portions. In addition, BMI was selected as a mediator to analyze the relationship between the frequency of EAFH and hypertension. The results of this study showed that BMI mediated and was partially mediating in the relationship between frequency of EAFH and hypertension, which was consistent with a study on the relationship between EAFH and diabetes [16].

We used the data from the Henan Rural Cohort Study to explore the relationship between the frequency of EAFH and hypertension. The study had a relatively large research population, used effective and validated questionnaires, had a high response rate, and conducted comprehensive data analysis, all of which were also strengths of this study. In addition, the blood pressure measurements of each participant were taken by trained investigators using the average of three measurements, which decreased the human error of blood pressure measurements.

This study had several shortcomings. First, since this was a cross-sectional study, the causal relationship between the frequency of EAFH and hypertension cannot be ascertained. Long-term research to obtain longitudinal data is still needed to verify this relationship. Furthermore, it was not feasible to measure the frequency of EAFH in the past week for each participant. Thus, the data on the frequency of this study were obtained by recollection, and participants’ unclear recollections of past exposures caused recall bias. Second, detailed information on dining out was not collected. For example, the places of EAFH were not recorded. Food from work or school canteens is different from that from restaurants or food stands, and the former may be healthier. Furthermore, whether one is alone or accompanied by others, as well as one’s mood, when EAFH affects the kinds and the amount of food consumed, further affecting the possibility of suffering from hypertension. Third, although more comprehensive covariates were included in this study, there are still some potential confounding factors that may modify the relationship between the frequency of EAFH and hypertension. Furthermore, this study concluded the effect of EAFH frequency on blood pressure and hypertension, but the specific nutritional components that produced this result were not explored. Therefore, further research is needed to confirm this relationship in the future.

## 5. Conclusions

This study showed that the higher the frequency of EAFH was, the higher the levels of SBP and DBP, and the possibility of hypertension, were, especially for Chinese rural men. In addition, there was a nonlinear dose–response relationship between the frequency of EAFH and hypertension, and BMI partially mediated this relationship. These results suggested that EAFH frequently may have adverse health outcomes. Therefore, it is of great necessity to implement relevant health policies, strengthen education on people’s healthy dietary habits and lifestyle, and provide proper training for catering practitioners to reduce the adverse effects of EAFH on health, especially in rural areas.

## Figures and Tables

**Figure 1 nutrients-14-01832-f001:**
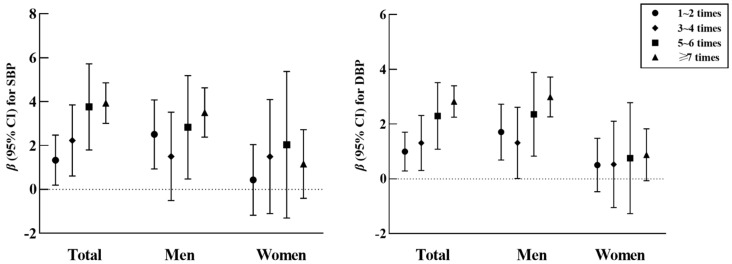
Association between frequency of EAFH and blood pressure. *CI*, confidence interval; SBP, systemic blood pressure; DBP, diastolic blood pressure; EAFH, eating away from home. Adjusted for age, gender (only for total participants), marital status, education level, average income per month, smoking status, alcohol consumption, physical activity, abundant vegetable and fruit intake, high-fat diet, and family history of hypertension.

**Figure 2 nutrients-14-01832-f002:**
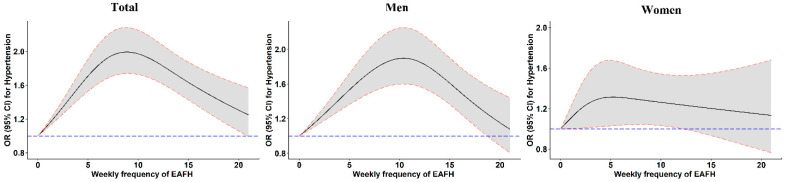
*ORs* and 95*% CIs* of weekly frequency of EAFH for hypertension from restricted cubic splines. *OR*, odds ratio; *CI*, confidence interval; EAFH, eating away from home. The fully adjusted model included age, gender (only for total participants), marital status, education level, average income per month, smoking status, alcohol consumption, physical activity, abundant vegetable and fruit intake, high-fat diet, and family history of hypertension.

**Figure 3 nutrients-14-01832-f003:**
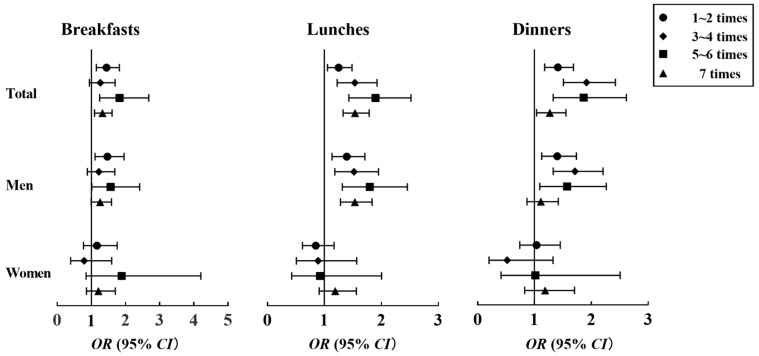
Associations between frequency of EAFH breakfasts, lunches, and dinners and hypertension among participants. *OR*, odds ratio; *CI*, confidence interval; EAFH, eating away from home. Adjusted for age, gender (only for total participants), marital status, education level, average income per month, smoking status, alcohol consumption, physical activity, abundant vegetable and fruit intake, high-fat diet, and family history of hypertension.

**Figure 4 nutrients-14-01832-f004:**
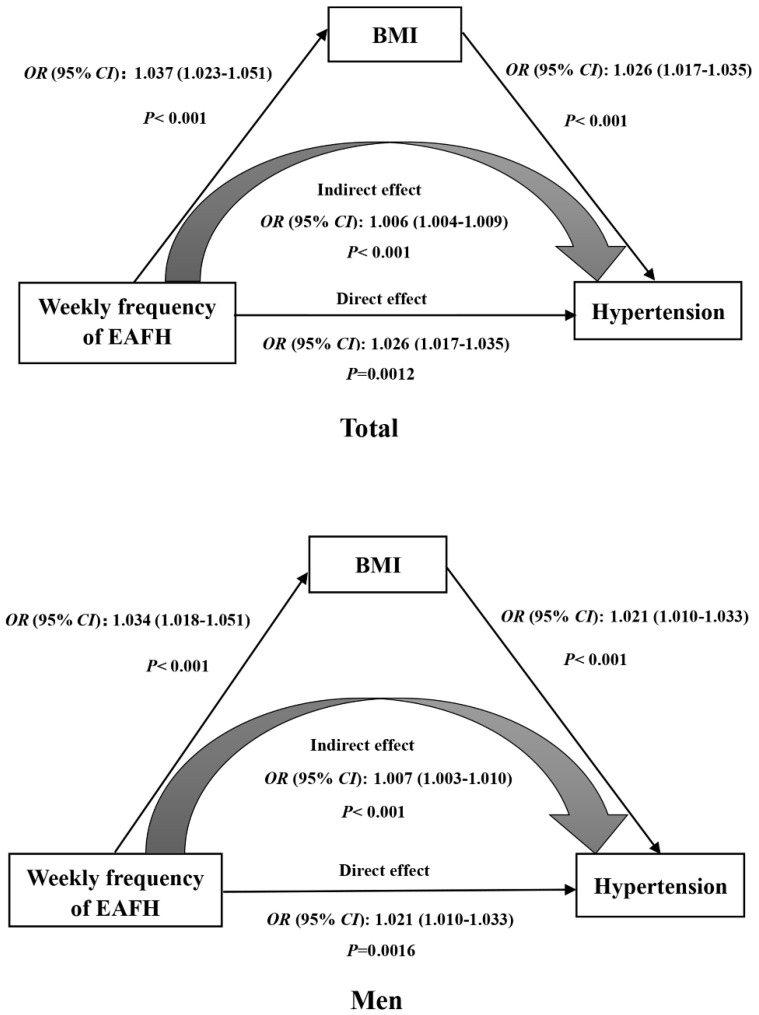
Mediation analysis of the relationship between the frequency of EAFH and hyper-tension by BMI for the total population and men. BMI body mass index; CI confidence interval; OR odds ratio. Adjusted for age, gender (only for total participants), marital status, education level, average income per month, smoking status, alcohol consumption, physical activity, abundant vegetable and fruit intake, high fat diet and family history of hypertension.

**Table 1 nutrients-14-01832-t001:** The characteristics of participants by gender.

Variables	Total (*N* = 29,611)			Men (*N* = 12,098)			Women (*N* = 17,513)		
	Normotensive	Hypertension	*p*	Normotensive	Hypertension	*p*	Normotensive	Hypertension	*p*
Age (years), mean ± SD	53.0 ± 12.6	60.5 ± 10.2	<0.001	54.9 ± 12.7	59.6 ± 11.1	<0.001	51.7 ± 12.4	61.2 ± 9.4	<0.001
Marital status, n (%)			<0.001			0.978			<0.001
Married/cohabitation	18,372 (91.3)	8314 (87.7)		7404 (90.2)	3511 (90.2)		10,968 (92.0)	4803 (85.9)	
Unmarried/divorced/widowed	1754 (8.7)	1171 (12.3)		802 (9.8)	381 (9.8)		952 (8.0)	790 (14.1)	
Education level, n (%)			<0.001			0.002			<0.001
Junior high school or below	16,426 (81.6)	8191 (86.4)		6440 (78.5)	3052 (78.4)		9986 (83.8)	5139 (91.9)	
High school	2831 (14.1)	1147 (12.1)		1423 (17.3)	724 (18.6)		1408 (11.8)	423 (7.5)	
High school above	869 (4.3)	147 (1.5)		343 (4.2)	116 (3.0)		526 (4.4)	31 (0.6)	
Average income per month, n (%)			<0.001			0.005			<0.001
<500 CNY	6882 (34.2)	3791 (40.0)		2941 (35.8)	1474 (37.9)		3941 (33.0)	2317 (41.4)	
500–1000 CNY	6296 (31.3)	3032 (32.0)		2486 (30.3)	1213 (31.2)		3810 (32.0)	1819 (32.5)	
≥1000 CNY	6948 (34.5)	2662 (28.0)		2779 (33.9)	1205 (31.0)		4169 (35.0)	1457 (26.1)	
Smoking status, n (%)			<0.001			<0.001			0.944
Never	14,394 (71.5)	6898 (72.7)		2519 (30.7)	1328 (34.2)		11,875 (99.6)	5570 (99.6)	
Former	1386 (6.9)	949 (10.0)		1374 (16.7)	943 (24.2)		12 (0.1)	6 (0.1)	
Current	4346 (21.6)	1638 (17.3)		4313 (52.6)	1621 (41.6)		33 (0.3)	17 (0.3)	
Alcohol consumption, n (%)			<0.001			<0.001			<0.001
Never	15,698 (78.0)	7272 (76.7)		4078 (49.7)	1756 (45.1)		11,620 (97.5)	5516 (98.6)	
Former	913 (4.5)	543 (5.7)		886 (10.8)	531 (13.6)		27 (0.2)	12 (0.2)	
Current	3515 (17.5)	1670 (17.6)		3242 (39.5)	1605 (41.3)		273 (2.3)	65 (1.2)	
Physical activity, n (%)			<0.001			<0.001			<0.001
Low	5920 (29.4)	3507 (37.0)		2592 (31.6)	1595 (41.0)		3328 (27.9)	1912 (34.2)	
Moderate	7774 (38.6)	3108 (32.7)		2403 (29.3)	996 (25.6)		5371 (45.1)	2112 (37.8)	
High	6432 (32.0)	2870 (30.3)		3211 (39.1)	1301 (33.4)		3221 (27.0)	1569 (28.0)	
Abundant vegetable and fruit intake, n (%)	10,239 (50.9)	3929 (41.4)	<0.001	4171 (50.8)	1624 (41.7)	<0.001	6068 (50.9)	2305 (41.2)	<0.001
High-fat diet, n (%)	4028 (20.0)	1362 (14.4)	<0.001	2075 (25.3)	802 (20.6)	<0.001	1953 (16.4)	560 (10.0)	<0.001
Family history of hypertension, n (%)	2966 (14.7)	2607 (27.5)	<0.001	1027 (12.5)	1051 (27.0)	<0.001	1939 (16.3)	1556 (27.8)	<0.001
BMI (kg/m^2^), mean ± SD	24.1 ± 3.4	26.0 ± 3.6	<0.001	23.9 ± 3.3	25.8 ± 3.5	<0.001	24.3 ± 3.4	26.1 ± 3.7	<0.001
Frequency of EAFH (times/week), n (%)			<0.001			0.492			<0.001
0	17,423 (86.6)	8512 (89.7)		6656 (81.1)	3169 (81.4)		10,767 (90.3)	5343 (95.5)	
1–2	794 (3.9)	257 (2.7)		361 (4.4)	162 (4.2)		433 (3.6)	95 (1.7)	
3–4	397 (2.0)	116 (1.2)		224 (2.7)	93 (2.4)		173 (1.5)	23 (0.4)	
5–6	261 (1.3)	84 (0.9)		162 (2.0)	66 (1.7)		99 (0.8)	18 (0.3)	
≥7	1251 (6.2)	516 (5.5)		803 (9.8)	402 (10.3)		448 (3.8)	114 (2.1)	
SBP (mmHg), mean ± SD	115.7 ± 11.9	146.9 ± 16.9	<0.001	117.2 ± 11.2	145.6 ± 16.1	<0.001	114.6 ± 12.3	147.8 ± 17.4	<0.001
DBP (mmHg), mean ± SD	72.4 ± 8.2	88.1 ± 10.7	<0.001	73.3 ± 8.3	89.5 ± 10.9	<0.001	71.8 ± 8.0	87.1 ± 10.4	<0.001

Continuous variables are represented by means and standard deviations, and categorical variables are expressed as percentages. SD, standard deviation; BMI, body mass index; WC, waist circumference; CNY, Chinese Yuan; SBP, systemic blood pressure; DBP, diastolic blood pressure; EAFH, eating away from home.

**Table 2 nutrients-14-01832-t002:** Multivariate-adjusted *OR*s and 95% *CI*s for hypertension according to the weekly frequency of EAFH.

Weekly Frequency of EAFH	Prevalence, % (95% *CI*)	*OR* (95% *CI*)				*P trend*
		Model 1	Model 2	Model 3	* Per Level Risk	
Total (n = 29,611)					1.031 (1.022–1.040)	<0.001
0 time (n = 25,935)	32.82 (32.25–33.39)	1 (Ref.)	1 (Ref.)	1 (Ref.)		
1–2 times (n = 1051)	24.45 (21.85–27.06)	1.147 (0.986–1.334)	1.173 (1.007–1.366)	1.115 (0.953–1.305)		
3–4 times (n = 513)	22.61 (18.98–26.24)	1.310 (1.051–1.632)	1.366 (1.094–1.704)	1.341 (1.067–1.684)		
5–6 times (n = 345)	24.35 (19.80–28.90)	1.545 (1.190–2.005)	1.625 (1.250–2.112)	1.576 (1.202–2.065)		
≥7 times (n = 1767)	29.20 (27.08–31.32)	1.645 (1.468–1.844)	1.723 (1.533–1.935)	1.673 (1.482–1.889)		
Men (n = 12,098)					1.026 (1.016–1.037)	<0.001
0 time (n = 9825)	32.25 (31.33–33.18)	1 (Ref.)	1 (Ref.)	1 (Ref.)		
1–2 times (n = 523)	30.98 (27.00–34.95)	1.307 (1.074–1.591)	1.291 (1.061–1.573)	1.190 (0.969–1.462)		
3–4 times (n = 317)	29.34 (24.30–34.38)	1.437 (1.114–1.855)	1.411 (1.093–1.823)	1.405 (1.077–1.833)		
5–6 times (n = 228)	28.95 (23.02–34.88)	1.514 (1.122–2.044)	1.475 (1.092–1.993)	1.419 (1.037–1.941)		
≥7 times (n = 1205)	33.36 (30.70–36.03)	1.693 (1.475–1.943)	1.653 (1.439–1.898)	1.634 (1.413–1.890)		
Women (n = 17,513)					1.010 (0.993–1.027)	0.245
0 time (n = 16,110)	33.17 (32.44–33.89)	1 (Ref.)	1 (Ref.)	1 (Ref.)		
1–2 times (n = 528)	17.99 (14.71–21.28)	0.976 (0.766–1.244)	0.992 (0.777–1.265)	1.006 (0.783–1.293)		
3–4 times (n = 196)	11.73 (7.19–16.28)	0.854 (0.540–1.353)	0.891 (0.561–1.417)	0.885 (0.551–1.422)		
5–6 times (n = 117)	15.38 (8.75–22.02)	1.183 (0.691–2.025)	1.274 (0.740–2.195)	1.393 (0.800–2.426)		
≥7 times (n = 562)	20.28 (16.95–23.62)	1.172 (0.936–1.467)	1.199 (0.956–1.503)	1.170 (0.926–1.478)		

*OR*, odds ratio; *CI*, confidence interval; EAFH, eating away from home. * Fully adjusted model including age, gender (only for total participants), marital status, education level, average income per month, smoking status, alcohol consumption, physical activity, abundant vegetable and fruit intake, high-fat diet, and family history of hypertension. Model 1: adjusted for age; Model 2: adjusted for age, gender (only for total participants), marital status, education level, and average income per month; Model 3: adjusted for age, gender (only for total participants), marital status, education level, average income per month, smoking status, alcohol consumption, physical activity, abundant vegetable and fruit intake, high-fat diet, and family history of hypertension.

## Data Availability

The data are available from the corresponding author on reasonable request. Contact Chongjian Wang (tjwcj2008@zzu.edu.cn) for additional information regarding data access.

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
