# Peer review of "Body Mass Index Mediates the Relationship between the Frequency of Eating Away from Home and Hypertension in Rural Adults: A Large-Scale Cross-Sectional Study"

_nutrients, 2022, doi:10.3390/nu14091832_

Round 1
Reviewer 1 Report
Body mass index mediates the relationship between the frequency of eating away from home and hypertension in rural adults: A large-scale cross-sectional study
General comments
The topic of the manuscript is very interesting, considering relationship between the fre- quency of eating away. Generally speaking, the article is well written, but I have some questions that need to be answered or explained especially to first part.
Introduction
Line 30-33
From 2000 to 2010, the incidence of hypertension increased 7.7% in low- and middle-income countries. China's nutrition and health survey illustrated that in 2002, 18% (153 million) of adults had hypertension , and the incidence rate was 29.6% in 2014
Can authors explain why the data presented are so old? Please refer to more recent information
Materials and Methods
Line 67-68
The subjects were selected from the general population aged 18 to 79
Why this age division was used? Do the authors suspect a relationship between the frequency of eating away from home and hypertension in these young people? Has it been hypothesized that they will get heart disease earlier than for example their parents?
Line 107-108
physical activity was classified into low, moderate, and high
what frequency of physical activity was included in this classification. How many minutes per day, per week? Was it specified exactly what type of activity it was: walking, training at the gym?
Results and discussion
General comment
I have no comments on the method of research analysis and discussion
Conclusions
In my opinion the topic is very interesting, In the future in this type of research it would be better to use a professional body mass analyzer. then you can compare the correlation of fat content with eating out and hypertension
Reviewer 2 Report
In this work, Liu and colleagues aimed to address the association between eating away from home and hypertension and whether this association is impacted by body mass index. Despite of the general interest of this manuscript, there are some major aspects that should still be considered.
General comment
- non-communicable diseases have raised in prevalence in the last years and intense research has been done in this matter. In this work, the authors have decided to pay a key focus in hypertension, but diabetes mellitus have increased equally and is also directly linked to dietary habits. Taking this into account, what is the novelty of this work over the already existing ones? Also, what was the reason for just selecting hypertension as the main disease?
Specific comments
- l. 37-42: revise this sentence, it is poorly constructed
- l. 46-49: there is no more recent data on such field? Please revise
- l. 80: please standardize the use of numbers in full or Arabic ones. Check the whole document
- l. 86: revise “without the use of anti-hypertensive drugs”
- l. 90-91: In this study, EAFH also included meals for take away? There is a lot of persons that buy meals to be eaten at home. Please clarify
- l. 107-112: how have you measured that?
- l. 141-143: revise this sentence
- l. 146-147: EAFH
- Table 1: how was normal distribution of data assessed? How can parametric tests be used with SD values higher than the mean? Please revise all statistical analysis
- Figure 1: statistical analysis was not performed here. Please revise
- Table 3: based on models presented, EAFH has no significant impact is women (model 1). How have you assessed in models 2 and 3 data stratified by gender if gender differences were stablished since the first analysis? First column of the table split data by gender. Please revise all data analysis
- Figure 4: looking at ORs presented, in the mediation analysis, the relationship between the frequency of EAFH and hypertension seems to be mainly of protection. Data presented here should be carefully revised
- l. 241: stratification by subregions were not performed in this study. Do you think important data if such layer is stablished?
- l. 258: fast food consumed at home are also considered?
- l. 268-274: currently, this is not a direct observation. Consumers are increasingly aware of the importance of controlling the intake of such macro and micronutrients
- l. 283: more than calories, it is important to know from where calories came from.
- l. 293: estrogens have a double action; only their positive effects are listed, but the negative ones should also be briefly underlined.
- l. 303-304: most women > please add percentage
- l. 313-320: and what about metabolic diseases?
- l. 333: a 7-day recall was applied? Are you aware of the bias introduced here?
- l. 349-357: people will always need for EAFH. The most important aspects to underline here is to ensure a proper educational level to the population on healthy dietary habits and lifestyle through implementing proper health policies in this field; similarly, proper training should be ensure to the staff working in restaurants, etc.
Round 2
Reviewer 2 Report
Accept